# Characterization of Meat Metabolites and Lipids in Shanghai Local Pig Breeds Revealed by LC–MS-Based Method

**DOI:** 10.3390/foods13152327

**Published:** 2024-07-24

**Authors:** Jun Gao, Lingwei Sun, Weilong Tu, Mengqian Cao, Shushan Zhang, Jiehuan Xu, Mengqian He, Defu Zhang, Jianjun Dai, Xiao Wu, Caifeng Wu

**Affiliations:** 1Institute of Animal Husbandry and Veterinary Science, Shanghai Academy of Agricultural Sciences, Shanghai 201106, China; gaojun@saas.sh.cn (J.G.); sunlingwei1987@126.com (L.S.); tuweilong@saas.sh.cn (W.T.); smalltreexj@126.com (S.Z.); jiehuanxu810@163.com (J.X.); he1037247863@163.com (M.H.); zhangdefuzdf@163.com (D.Z.); daijianjun@saas.sh.cn (J.D.); 2Key Laboratory of Livestock and Poultry Resources (Pig) Evaluation and Utilization, Ministry of Agriculture and Rural Affairs, Shanghai 201106, China; 3Shanghai Municipal Key Laboratory of Agri-Genetics and Breeding, Shanghai 201106, China; 13887223118@163.com; 4Shanghai Engineering Research Center of Breeding Pig, Shanghai 201106, China; 5College of Fisheries and Life Science, Shanghai Ocean University, Shanghai 201306, China; 6Biotechnology Research Institute, Shanghai Academy of Agriculture Sciences, Shanghai 201106, China

**Keywords:** metabolomics, lipidomics, liquid chromatography–mass spectrometry, longissimus dorsi muscle, gluteus muscle

## Abstract

The meat of local livestock breeds often has unique qualities and flavors. In this study, three Shanghai native pig breeds (MSZ, SWT, and SHB) exhibited better meat quality traits than globalized commercial pig breeds (DLY). Subsequently, metabolomic and lipidomic differences in the longissimus dorsi (L) and gluteus (T) muscles of the Shanghai native pig breeds and DLY pig breed were compared using liquid chromatography–mass spectrometry (LC–MS). The results demonstrated that the metabolites mainly consisted of (28.16%) lipids and lipid-like molecules, and (25.87%) organic acids and their derivatives were the two most dominant groups. Hundreds of differential expression metabolites were identified in every compared group, respectively. One-way ANOVA was applied to test the significance between multiple groups. Among the 20 most abundant differential metabolites, L-carnitine was significantly different in the muscles of the four pig breeds (*p*-value = 7.322 × 10^−11^). It was significantly higher in the L and T muscles of the two indigenous black pig breeds (MSZ and SWT) than in the DLY pigs (*p*-value < 0.001). Similarly, lipidomic analysis revealed the PA (18:0/18:2) was significantly more abundant in the muscle of these two black breeds than that in the DLY breed (*p*-value < 0.001). These specific metabolites and lipids might influence the meat quality and taste properties and lead to customer preferences. Therefore, this study provided insights into the characterization of meat metabolites and lipids in Shanghai native pig breeds.

## 1. Introduction

Livestock and poultry products are an important source of protein in the human diet. Among them, pork, as one of the most important sources of meat, plays a pivotal role in residential meat consumption. China is a major pork producer and consumer in the world, and it is reported that China’s pork production has reached 52.96 million tons in 2020, accounting for more than 59.6% of meat production [1]. The Duroc × Landrace × Yorkshire crossbreed (DLY) has the biggest market share in China and is characterized by high leanness, fast growth, and high feed conversion efficiency [2,3]. However, the meat of the DLY breed has relatively inferior quality characteristics that include PSE (pale, soft, and exudative) or DFD (dark, firm, dry), which tend to cause consumer dissatisfaction [4,5]. Genetic defects and pre-slaughter stress have been identified as important factors [6,7]. It has been shown that myocytes lose water, and the reflectance of incident light decreases, which can lead to paleness [6]. Alternatively, when exposed to high temperatures, muscle protein denaturation is accelerated, leading to paleness [8,9] and reduced water holding capacity (WHC) [9,10]. Liu et al. applied metabolomics analysis to study the difference in meat quality between the Chuanzang black (CB) pig and DLY breed [11].

Chinese native pig breeds have their own unique characteristics, such as a higher intramuscular fat content (IMF) and better meat quality. It might be formed by long-term artificial or natural selection [3,5]. The content and composition of free amino acids and the fatty acid in the muscle play an important role in determining the nutritional profile and directly affect its taste properties [12,13,14]. Metabolomics and lipidomics can provide more comprehensive insights of the metabolites and lipids of the meat [3]. LC–MS technology has been widely used in the metabolomic and lipidomic detection of pork because of its high sensitivity and wide detection range [15,16], such as the comparative study on PSE and red, firm, and non-exudative pork [9], and the identification of metabolomic changes in longissimus dorsi muscle of Finishing pigs following heat stress [17]. With the improvement of global living standards, meat quality has become a key factor influencing consumers’ purchasing decisions [18]. However, the profiles of meat metabolomics and lipidomics of Shanghai native pig breeds have not been systematically elucidated. Therefore, through muscle untargeted metabolomics and lipidomics among Meishan Pig (MSZ), ShaWutou Pig (SWT), Shanghai White breed (SHB) [19], and commercial DLY breed, this study aimed to dissect differential metabolic biomarkers to characterize Shanghai local pig breeds in terms of meat quality, which will provide novel insights for the development and utilization of Shanghai local pig breeds.

## 2. Materials and Methods

### 2.1. Sample Collection

A total of three Shanghai native pig breeds (MSZ, SWT, and SHB) and one commercial pig breed (DLY) were selected for this study. MSZ was from Meishan Pig Breeding Center, Jiading District, Shanghai; SWT was from Shawutou Breeding Farm, Chongming District, Shanghai; and SHB was from Breeding Farm of Zhuanghang Comprehensive Experimental Base of Shanghai Academy of Agricultural Sciences (SAAS). DLY was from Wufeng Shangshi Foods Co. (Shanghai, China). Appearance pictures and feed nutrient levels of the pig breeds were supplemented in Appendix A. The pork samples were collected from these 4 breeds that had reached the standardized market slaughter weight (100 ± 5 kg). All slaughtering was performed by licensed commercial slaughtering companies, such as Wufeng Shangshi Food Co., and the researchers only handled the associated meat samples. Twelve heads of each breed, half male and half female, were collected from individual with two tissue parts including the longissimus dorsi muscle (L) and gluteal muscle (T). A total of 96 pork samples were obtained.

### 2.2. Measurement of Meat Quality Traits

Measurements were taken of 10 individuals per breed. Samples were taken from the thoracolumbar junction to the L muscle of the 3rd and 4th lumbar vertebrae for the measurement of meat quality traits. The meat sample was taken and quickly put into a self-sealing plastic bag, stored in a 4 °C refrigerator, and brought back to the laboratory for testing. Petroleum ether and other reagents required for the experiment were purchased from Sinopharm Chemical Reagent Co. (Shanghai, China). Meat quality traits, such as intramuscular fat (IMF), 24-h drip loss, meat color, and water holding capacity (WHC), were tested. Measurements were made with reference to the reported methods [11,20], and the instruments used were a pH meter (Thermo Star, Thermo Fisher Scientific, Waltham, MA, USA), color difference analyzer (SP60, X-rite, Grand Rapids, MI, USA), automatic fat analyzer (XT10, ANKOM, Macedon, NY, USA), and tenderness meter (C-LM3, Beijing Bullard Technology Development Co., Ltd., Beijing, China).

### 2.3. Metabolite and Lipid Extraction from Tissues

This study used 6 mm grinding beads to extract the metabolites from 50 mg of skinless muscle frozen tissue. An extraction solution of 400 μL methanol to water in a volume ratio of 4:1 was used. L-2-chlorophenylalanine (0.02 mg/mL) was added as an internal standard. Lipid extraction used 280 µL of methanol solution (2:5 volume ratio with water) and 400 µL of methyl tertiary butyl ether. The extract was ground for 6 min at 50 Hz at −10 °C, followed by low temperature extraction at 5 °C for 30 min (40 kHz), followed by standing at −20 °C for 30 min and centrifugation at 4 °C for 15 min (13,000× *g*).

### 2.4. LC–MS of Metabolites

The system used for the LC–MS analysis was a Thermo UHPLC-Q Exactive HF-X with an ACQUITY HSS T3 column (Waters, Milford, CT, USA). The mobile phases were 0.1% formic acid water: acetonitrile (volume ratio is 95:5) and 0.1% formic acid acetonitrile: isopropanol: water (volume ratio is 47.5:47.5). The column temperature was set at 40 °C, the flow rate was set at 0.40 mL/min, and the mass spectral signals of the samples were collected in positive and negative ion scanning modes. Quality control (QC) samples were prepared by mixing all samples in equal volumes. The experimental part of LC–MS was carried out by Shanghai Majorbio Company (Shanghai, China) (https://www.majorbio.com/ (accessed on 8 September 2023)) according to the established procedure [21,22].

### 2.5. LC–MS of Lipids

The same LC–MS analysis system as described above was used and equipped with an Accucore C30 column (Thermo Scientific, Waltham, MA, USA). The mobile phases consisted of 10 mM ammonium acetate in acetonitrile: H_2_O (volume ratio is 1:1) (0.1% formic acid) and 2 mM ammonium acetate in acetonitrile: isopropanol: H_2_O (volume ratio is 10:88:2) (0.02% formic acid). The standard sample injection parameters were as follows: 2 µL volume, flow rate set at 0.4 mL/min, column temperature set at 40 °C, and 20 min of total chromatographic separation. The QC samples were prepared as described previously.

### 2.6. Data Analysis

The LC–MS raw data were converted into the common format by Progenesis QI software 2.3 (Waters, Milford, CT, USA) [23] and Lipidsearch 4.2 (Thermo Fisher, San Diego, CA, USA). The metabolites were identified by searching the HMDB database (http://www.hmdb.ca/ (accessed on 12 September 2023)) [24] and Metlin (https://metlin.scripps.edu/ (accessed on 12 September 2023)) [25]. Metabolic and lipidomic features detected in at least 80% in any set of samples were retained [26,27]. Variables with a relative standard deviation (RSD) more than 30% of the QC samples were removed, and log10 logarithmization was performed [28]. The data were analyzed through the free online platform (https://Cloud.majorbio.com/ (accessed on 13 September 2023)) [29,30].

Variance analysis was performed on the matrix file after data preprocessing. Overall differences among the groups were analyzed by PLS-DA (partial least squares discriminant analysis) [31]. The number of times for the randomized permutation test was set to 200. The orthogonal least partial squares discriminant analysis (OPLS-DA) and 7-cycle interactive validation were used to evaluate the stability of the model. In addition, variable importance in the projection (VIP) [32] analysis obtained by the OPLS-DA model and the *p*-value of the Student’s *t*-test, and the different expression metabolites (DEMs) with VIP score > 1 and *p*-value < 0.05 were considered as significant.

Venn plot (Venn) [33] is used for plots that show overlap of the number of the elements. One-way ANOVA [34] was used to compare the distribution of metabolites in three or more sample groups for significant differences, and then post hoc tests [35] compared two-by-two to detect sample groups that differed in multiple groups. Student’s *t*-test [36] (unpaired) was used in the comparison of differential metabolites between Shanghai local pig breeds and DLY pig breeds, respectively. By default, the program screened for significant differences based on a *p*-value < 0.05.

Differential metabolites were mapped into their biochemical pathways based on a KEGG database search (http://www.genome.jp/kegg/ (accessed on 13 September 2023)) [37].

## 3. Results

### 3.1. Meat Quality Traits

This study compared differences in meat quality traits among three Shanghai local pig breeds and the commercial DLY breed. In Table 1, based on the *p*-value of the one-way ANOVA comparison, the results clearly indicate that, except for water content and shear force, which were not significantly different among the four breeds, all other meat quality traits were significantly different among the four breeds (*p* < 0.05). Intramuscular fat (IMF), protein content, lightness (L*), and yellowness (b*) were significantly higher in Shanghai local pigs than in the commercial DLY breed.

The Pearson correlation results on these meat traits demonstrated that the protein content (%) showed a significant negative correlation with water content (%) (r = −0.51, *p*-value = 1.59 × 10^−4^. The meat lightness (L) trait showed a significant positive correlation with the yellowness (b) trait (r = 0.62, *p*-value = 1.785 × 10^−6^), protein content % (r = 0.45, *p*-value = 0.001), and IMF% (r = 0.44, *p*-value = 0.001).

### 3.2. Muscle Untargeted Metabolomic Analysis

A total of 786 (pos) and 343 (neg) metabolites were identified in this study. Significant differences are observed in the metabolites of interest, both when comparing the breeds and comparing the different parts of the pork (Figure 1A); it can be clearly observed that the metabolites of the samples from both MSZ (L and T) and DLY (L and T muscle) are clustered together individually. The permutation test model is to randomly disrupt the grouping labels (Y variables) of the experimental and control groups. The intercept of the Q2 regression line with the *y*-axis is less than 0.05 (Figure 1B), indicating that the model is robust and not overfitted. The results demonstrated a significant difference between the different breeds or parts of meat.

### 3.3. DEMs Identified between Breeds

The classification of HMDB compounds revealed that DEMs are 283 (28.16%) lipids and lipid-like molecules and 260 (25.87%) organic acids and their derivatives, which are the most numerous two classes of compounds (Figure 2A). Hundreds of DEMs were identified through comparison, respectively. For example, 357 DEMs were identified in the comparison of the L muscles of the DLY and MSZ groups (Figure 2B) for the one-way ANOVA analysis, revealing the significance of the 20 differential metabolites with the highest abundance among groups (Figure 2C).

For example, the abundance of L-carnitine (*p*-value = 7.322 × 10^−11^) and inosine (7.985 × 10^−17^) differed significantly between groups. The abundance of L-carnitine in L and T meat tissues was significantly lower (*p*-value < 0.001) in the commercial pig breed (DLY) than in the two Shanghai indigenous black pig breeds (MSZ and SWT) (Figure 2D). Similarly, there was also significant variability for L-acetylcarnitine (*p*-value = 1.53 × 10^−9^). In contrast, the abundance of inosine in the two Shanghai indigenous pig breeds (MSZ and SWT) was significantly lower (*p*-value < 0.001) than that in the two white pig breeds (DLY and SHB) (Figure 2E). Detailed information on the DEM sets for each group was added in Appendix A.

Differential metabolite sets identified by comparing three Shanghai local pig breeds with the commercial pig breed DLY were analyzed by a Venn diagram. We identified 50 shared important differential metabolites in L muscle tissue (Figure 3A,B). It was found that the most significant differences remained for carnitines, such as decanoylcarnitine (Figure 3C), L-octanoylcarnitine (Figure 3D), and lauroylcarnitine (Figure 3E). Using the same method, we also analyzed T tissues and identified a total of 178 shared significant differential metabolites, among which 1-(beta-D-ribofuranosyl)-1,4-dihydronicotinamide (Figure 3F) had a significantly higher abundance in the T muscle of the commercial pig DLY than in the three local pig breeds in Shanghai. On the contrary, 10-formyldihydrofolate (Figure 3G) had a higher abundance in the T muscle of the Shanghai local pig breeds. On the other hand, 2-phenylethyl acetate had a lower abundance in the T muscle of the two indigenous black pig breeds (MSZ and SWT) in Shanghai than in the two white pig breeds (DLY and SHB) (Figure 3H).

### 3.4. DEMs Identified between L and T Meat Parts

In addition to comparing differences between breeds, this study also identified muscle differential metabolites in different parts of the same breed. By analyzing the DEMs of L and T muscles of each of the four breeds, 406, 290, 264, and 229 differential metabolites were identified in the DLY, SHB, MSZ, and SWT groups, respectively (Figure 4A). The Venn diagrams showed that there were 73 DEMs shared among them (Figure 4A), indicating that these 73 DEMs were significantly different metabolic collections between L and T muscle. We also performed a VIP analysis of these 73 DEMs and revealed the top 30 DEMs with the highest VIP scores (Figure 4B). The significant different abundance of the three highest-scoring metabolites, glutamylvaline, phenylacetylglutamine, and glucaric acid, were demonstrated among the groups (Figure 4C).

### 3.5. Lipidomic Analysis

Lipid metabolites can be categorized into different classes based on different classification methods to facilitate research for specific research directions. The lipid metabolites and pathways strategy [38] [project (LIPID MAPS)] classified lipid metabolites into eight major categories: fatty acids (FA), glycerolipids (GL), glycerophospholipids (GP), sphingolipids (SP), sterol lipids (ST), prenol lipids (PR), saccharolipids (SL), and polyketides (PK). A total of 483 (pos) and 273 (neg) lipid metabolites were identified in this study that consisted of four main categories (GP, GL, SP, and FA) (Figure 5A). An analysis of the KEGG pathway (Figure 5B) revealed that these differential expressed lipids (DELs) were mainly enriched in choline metabolism in cancer, fat digestion and absorption, and GnRH signaling pathways.

The Venn diagrams showed that 13 DELs were shared among the six groups in the comparison between Shanghai local and DLY pig breeds (Figure 5C). Most of them were mainly concentrated on glycerophospholipids (except GM3 belonged to Sphingolipids). Among them, VIP analysis showed that PEt (18:1e/20:4), PA (18:0/18:2), and GM3 (d16:0/20:1) had the highest VIP values in the differences between L muscles of MSZ and DLY (Figure 5D).

It is worth noting that the PA (18:0/18:2), chemical formula: C39H73O8P, is a phosphatidic acid and involved in several of the most prominent metabolism pathways mentioned above, such as fat digestion and absorption, glycerophospholipid metabolism, GnRH signaling pathway, etc. It is clearly shown that its abundance is significantly different (*p*-value = 6.207 × 10^−20^) among the four pig breeds. In particular, in the commercial pig breed DLY, it is significantly lower (all *p* < 0.001) than in the two black pig breeds (MSZ and SWT) (Figure 5E).

### 3.6. DELs Identified between L and T Meat Parts

The present study also identified muscle DELs in different meat tissues of the same breed by analyzing the DELs of L and T muscles in each breed. A total of 162, 128, 173, and 152 DELs were identified in the DLY, SHB, MSZ, and SWT groups, respectively (Figure 6A). On the other hand, the Venn diagrams showed 17 shared DELs between L and T muscles, respectively (Figure 6A). The VIP analysis of these 17 DELs presents those with the highest VIP scores (Figure 6B) and presents the abundance of the three highest scoring lipids: AcCa (22:0), CL (22:2/18:0/18:1/20:4), and BisMePA (18:0/20:5) (Figure 6C). Fifteen of these 17 DELs were also predominantly distributed in glycerophospholipids, while the other two, AcCa (22:0) and Co (Q9), belonged to fatty acids (FA).

## 4. Discussion

Several studies have shown that Chinese indigenous pig breeds have relatively better meat quality (e.g., higher intermuscular fat content and better meat color) and flavor than the globalized hybrid commercial pig DLY, which is known for its growth rate, muscle deposition, and lean meat yield [5,39]. Similar results were also obtained in this study, for example, the IMF, protein content, and meat color of the Shanghai local breeds were significantly higher than those of the DLY breeds, indicating that the Shanghai local breeds also have relatively better meat quality. Among the 20 metabolic markers with the highest abundance in meat, we found that L-carnitine and L-acetylcarnitine were significantly higher in two meat tissues (L and T muscle) of two indigenous black pig breeds (MSZ and SWT) than in the commercial pig breed (DLY).

L-carnitine is an endogenous molecule involved in fatty acid metabolism and can be found in many foods, especially in red meat. L-carnitine transports fatty acid chains into the mitochondrial matrix, which allows for cells to break down fat and derive energy from stored fat reserves [40]. L-carnitine and its esters help reduce oxidative stress, exert a nutrigenomic effect via direct modulation of nuclear receptor signaling in adipocytes, demonstrating its nutritional impact, and have been proposed as treatments for many diseases [40,41]. The L-carnitine levels and storage stabilities of livestock products were reported to have a high correlation (r = 0.9764) with the redness values of a homogenized meat solution [42]. Gao et al. identified nine differential metabolites, including L-carnitine, that were involved in antioxidant function, lipid metabolism, and cell signal transduction, which may decrease postmortem meat quality and play important roles in anti-heat stress [17]. Comparisons of carcass traits, meat quality, and serum metabolomes between Chinese Shaziling pigs and Yorkshire pigs have also been reported, and higher serum L-carnitine levels have been found to be a promising indicator of improved meat quality [43]. From this point of view, the meat quality of Shanghai local pig breeds is also relatively better than that of commercial DLY breeds.

In a previous study, Piglets, especially those of low birth weight, may benefit from carnitine supplementation in the early postnatal period, which may mitigate the negative effects of low birth weight on body composition and market weight meat quality [44]. In addition, it has been shown that reconfiguration of carnitine metabolism using a Chinese obese Ningxiang pig-derived microbiota can promote muscle fatty acid deposition in lean DLY breed [45]. Therefore, the differences in carnitine levels between Shanghai local pig breeds and the DLY breed were confirmed in this study. Other distinctly different metabolites are also of interest, which may also have an important impact on meat flavor, such as the abundance content of inosine that was significantly lower in the two Shanghai indigenous black pig breeds than in the two white pig breeds (DLY and SHB). Inosine 5′-monophosphate (IMP) is an important metabolite that contributes to meat flavor [46]. The breakdown of inosine to form hypoxanthine along with free amino acids contributes to a bitter taste [47,48].

Differences in metabolites between different parts of pork may be an important factor contributing to variations in physicochemical properties, nutrient content, and flavor of different meats. Glutamylvaline is a peptide composed of glutamate and valine and is a proteolytic breakdown product of larger proteins. And there are reports that kokumi-active γ-glutamyl peptides could enhance the umami taste of monosodium glutamate [49]. Phenylacetylglutamine is a product of the conjugation of phenylacetic acid with glutamine. A study of dietary intake patterns demonstrated that *O*-acetylcarnitine and phenylacetylglutamine are positively correlated with the intake of red meat and vegetables, respectively [50]. The analysis of these metabolite differences helped to characterize the differences between L and T muscles at the level of small molecule metabolites.

A study comparing the lipidomics of the longest dorsal muscle of Chinese indigenous Luchuan pigs with that of DLY identified that the expression of 61 TGs and 20 DGs were significantly up-regulated in Luchuan pigs, and only three were down-regulated. The study concluded that this explains the better meat flavor of Luchuan pigs than Duroc pigs [15]. Similar findings were obtained in the present study, where GP (of which PC was the most abundant) and GL (including TG and DG) were the predominant differential lipids in terms of lipid group classification. In the comparison of the longissimus dorsi muscle of MSZ with those of DLY, GL lipids (TG+DG) generally showed an up-regulation in MSZ pigs. Both MSZ and Luchuan pigs belong to the local obese pig breeds in China, which are characterized by higher intermuscular fat content compared to the lean commercial pig DLY.

Through multi-group comparisons, we identified 13 DELs shared between the DLY breed with the Shanghai native breeds and 17 DELs between L and T muscles. Not many studies have been reported on these DELs. For example, PA (18:0/18:2) of the 13 DELs, chemical formula: C39H73O8P, is a phosphatidic acid. This study showed that its abundance level in meat was significantly lower in the DLY breed. And this lipid was involved in several lipid metabolic pathways. The major biosynthetic pathway of PA in animal tissues involves sequential acylation of α-glycerophosphate by acyl-CoA derivatives of fatty acids, making it a biologically active lipid that stimulates a variety of responses such as smooth muscle contraction. Few studies have been reported on this lipid, and in particular, its differences in meat quality among pig breeds have not been revealed. How it affects meat traits or flavor still remains to be elucidated.

## 5. Conclusions

In this study, we compared the meat quality traits of three Shanghai local pig breeds with the commercial crossbred pig breed DLY. IMF, protein content, and meat color traits reflect the better meat quality in Shanghai local pig breeds. Based on LC–MS analysis results, differences in the abundance of carnitine metabolites, such as L-carnitine, were the most significant muscle metabolism differentiating substances between Shanghai local pig breeds and DLY pig breeds. Characteristic differential lipids were also identified among the breeds, such as PA (18:0/18:2) abundance in the muscle of Shanghai local pigs, both of which were significantly higher than that of the DLY breed. Therefore, these muscle-related biomarkers identified in this study provided novel insights on characterization of pork metabolites in Shanghai local pig breeds.

## Figures and Tables

**Figure 1 foods-13-02327-f001:**
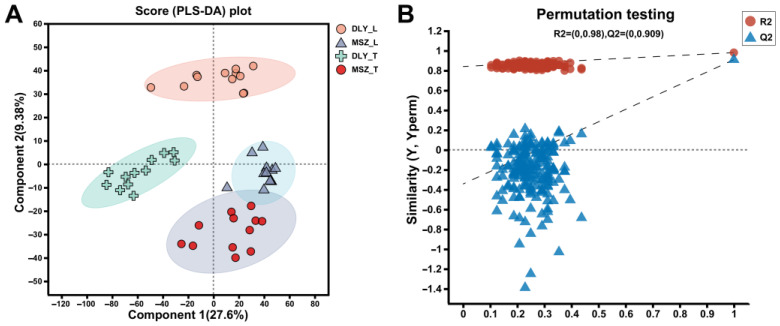
Partial least squares discriminant analysis (PLS-DA). (**A**) Metabolite PLS-DA plot of MSZ breed and DLY breed. The component 1 is the first principal component explanatory degree, and the component 2 is the second principal component explanatory degree. (**B**) Permutation test of PLS-DA model. The horizontal coordinates represent the replacement retention rate of the replacement test, the vertical coordinates represent the R2 (blue triangles) and Q2 (red dots) replacement test values, and the two dashed lines represent the regression lines for R2 and Q2, respectively.

**Figure 2 foods-13-02327-f002:**
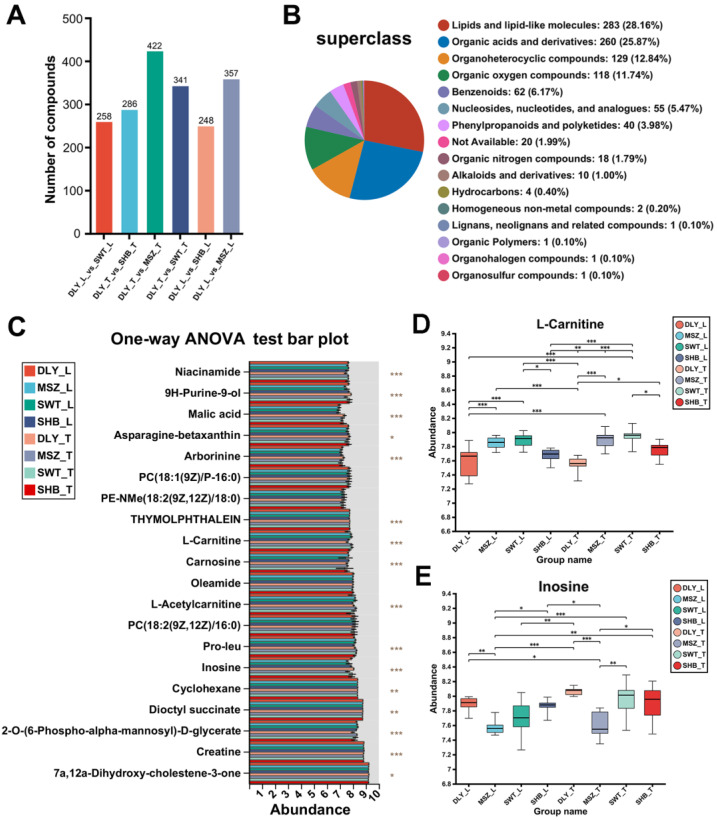
Differential metabolites between Shanghai local pig breeds and DLY breed. (**A**) Statistics of differential metabolites in L and T tissues of DLY versus three other breeds of pigs. (**B**) Compound classification statistics of differential metabolites using the Human Metabolome Database (HMDB). (**C**) One-way ANOVA analysis between groups and then post hoc tests. *y*-axis indicates the top 20 most abundant metabolite names, and *x*-axis indicates the average relative abundance of metabolites in different subgroups; *p*-value is shown on the rightmost side, * *p*-value < 0.05, ** 0.001 < *p* < 0.05, *** *p*-value < 0.001. (**D**) Abundance plot of L-carnitine between groups. (**E**) Abundance plot of inosine between groups.

**Figure 3 foods-13-02327-f003:**
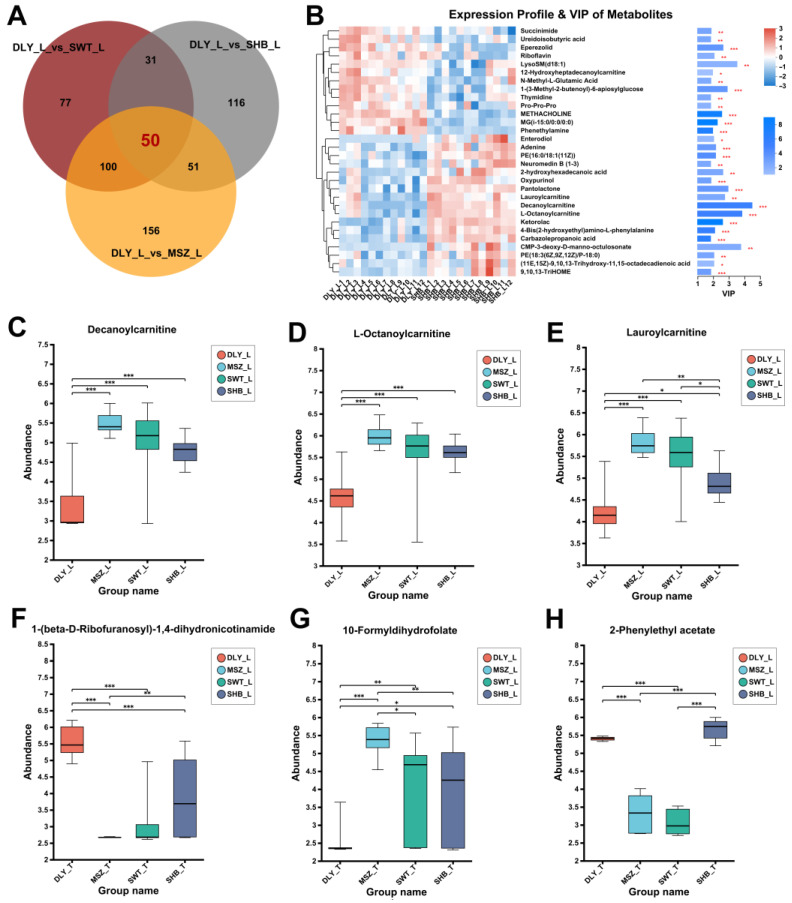
Shared DEMs between Shanghai local pig breeds and DLY breed. (**A**) Venn diagrams of the DEM sets identified between L groups. (**B**) VIP analysis of this collection of top 30 metabolites. On the left side is the metabolite clustering dendrogram, with the sample name at the bottom. On the right side, the bar length indicates the contribution of the metabolite to the difference. The bar color indicates the significance of the difference of the metabolite. The smaller the *p*-value, the larger −log10 (*p*-value), the darker the color. (**C**–**E**) correspond to the abundance of the three significantly different metabolites in the L groups, respectively. (**F**–**H**) correspond to the abundance of the three significantly different metabolites in the T groups, respectively. * means *p*-value < 0.05, ** means *p*-value < 0.01, and *** means *p*-value < 0.001.

**Figure 4 foods-13-02327-f004:**
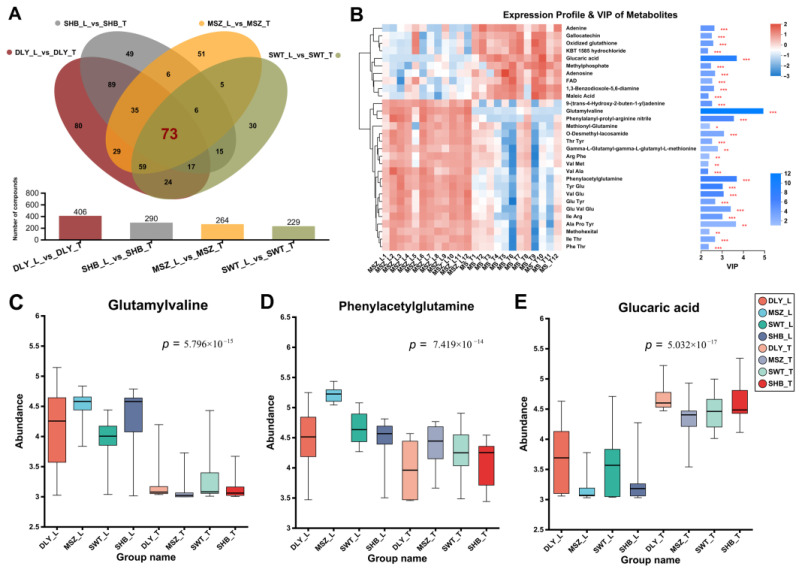
DEMs between longissimus dorsi (L) and gluteal muscle (T). (**A**) Venn diagrams of the metabolic sets identified in 4 breeds between L and T meat tissues. (**B**) Results of the VIP analysis of the 73 shared differential metabolite collections, with a graphical representation of the 30 DEMs with the top VIP values among them. (**C**–**E**) demonstrate the abundance of three DEMs with the top VIP values in each group. Significant *p*-values obtained from the ANOVA test are shown on the plot. * means *p*-value < 0.05, ** means *p*-value < 0.01, and *** means *p*-value < 0.001.

**Figure 5 foods-13-02327-f005:**
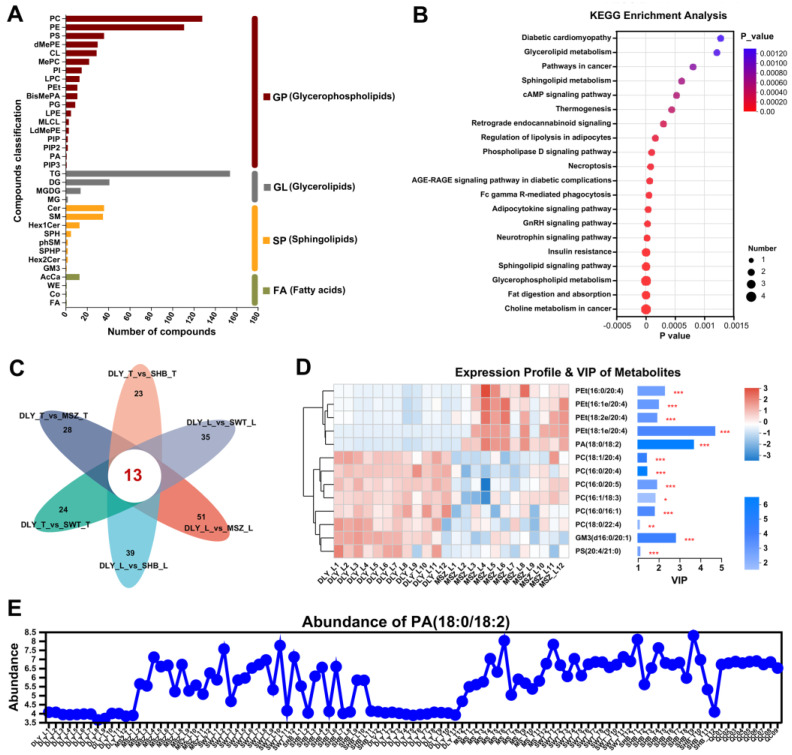
Lipid analysis of Shanghai local pig breeds compared with DLY breed. (**A**) Classification information of lipids in this study obtained based on LIPID MAPS. (**B**) Enrichment analysis of KEGG pathway for differentially metabolized lipids. (**C**) Venn diagrams of the metabolic sets identified in each group. (**D**) VIP analysis of 13 differential expressed lipids (DELs) shared by various groups. (**E**) The PA (18:0/18:2) abundance in all samples. * means *p*-value < 0.05, ** means *p*-value < 0.01, and *** means *p*-value < 0.001.

**Figure 6 foods-13-02327-f006:**
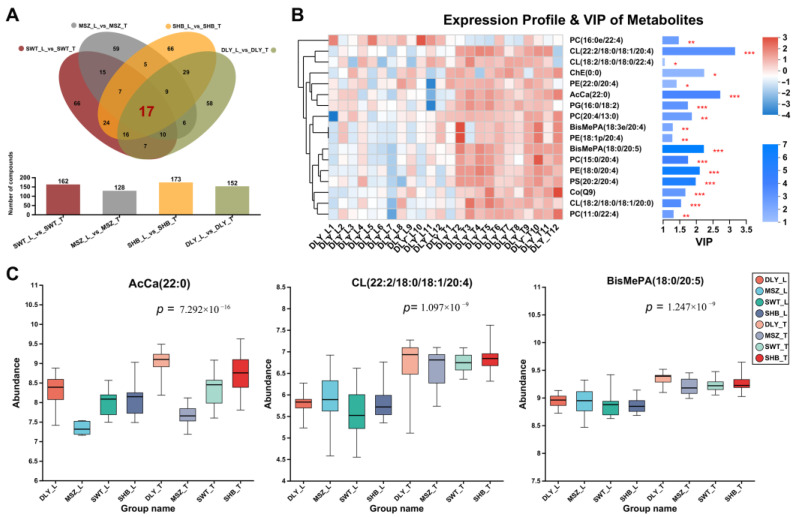
Differential analysis of lipids between L and T meat parts. (**A**) Venn diagrams of the differential expression lipids sets identified in 4 breeds between L and T type meat. (**B**) Results of the VIP analysis of the17 shared differential lipids collection. (**C**) Abundance of the three differential lipids with the highest VIP values in each group. * means *p*-value < 0.05, ** means *p*-value < 0.01, and *** means *p*-value < 0.001.

**Table 1 foods-13-02327-t001:** Results of measurement of meat quality traits.

Meat Quality Traits	DLY	MSZ	SWT	SHB	*p*-Value
IMF (%)	3.22 ± 0.77	5.77 ± 0.51	4.74 ± 1.06	4.42 ± 2.29	**
Drip loss (%)	4.76 ± 1.73	3.36 ± 1.09	5.93 ± 0.26	4.66 ± 1.79	**
pH	5.78 ± 0.30	6.46 ± 0.27	5.61 ± 0.29	5.63 ± 0.26	****
Lightness (L*)	15.48 ± 4.20	44.14 ± 2.88	50.21 ± 6.87	48.44 ± 6.56	****
Redness (a*)	11.00 ± 1.52	3.81 ± 2.32	8.35 ± 2.64	6.98 ± 3.38	*
Yellowness (b*)	6.20 ± 1.71	9.39 ± 1.48	9.37 ± 2.67	10.67 ± 1.04	****
Water content (%)	69.08 ± 1.47	69.79 ± 3.62	65.68 ± 6.28	67.50 ± 2.06	ns
WHC (%)	85.21 ± 4.28	91.86 ± 5.02	91.25 ± 6.73	89.65 ± 4.69	*
Protein content (%)	11.19 ± 1.79	14.42 ± 5.34	18.63 ± 2.94	13.10 ± 2.54	***
Shear force (kgf)	3.35 ± 0.74	2.57 ± 0.31	4.79 ± 2.13	3.65 ± 0.42	ns

Note: IMF represents intramuscular fat, and WHC represents water holding capacity. “ns” means that the difference is not significant (*p*-value > 0.05); “*” means that the *p*-value < 0.05, “**” means that the *p*-value < 0.01, “***” means that the *p*-value < 0.001), and “****” means that the *p*-value < 0.0001.

## Data Availability

The original contributions presented in the study are included in the article/Appendix A, further inquiries can be directed to the corresponding authors.

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
