# Peer review of "Characterization of Meat Metabolites and Lipids in Shanghai Local Pig Breeds Revealed by LC–MS-Based Method"

_foods, 2024, doi:10.3390/foods13152327_

Round 1

Reviewer 1 Report

Comments and Suggestions for Authors

The study is an interesting work on metabolomic biomarkers that influence the quality of various pig breeds. I believe that it has sufficient quality and scientific merit to be published in said journal, however, the authors must make some corrections in order to improve the manuscript.

1. Keywords must remove those that appear in the title such as: 'Shangai Local', 'Pig Breeds', LC-MS.

2. Introduction: in line 33, data must be entered worldwide and specifically in Shanghai on the production and consumption of this type of meat. In lines 42-43, the production levels or intervals of this type of compounds should be discussed in relation to the bromatological analysis: (ash, proteins, humidity, carbohydrates and fats). Some studies on the use of LC-MS for this type of matrices should be included in lines 46-47.

3. Materials and methods: Justify the experimental design used. Why are 96 samples used?

In the methodologies described for sections 2.3; 2.4 and 2.5 the authors must include the references of the methodologies they are using in case they are not theirs.

4. Results and discussion: In Table 1, it would be interesting to perform a Pearson correlation between the variables. The figures have to be improved, since the resolution is not appropriate. In addition, the names of the metabolites must be standardized according to the IUPAC, in addition, some appear in capital letters and others in lower case.

5. Discussion: the authors should reinforce the discussion by justifying how metabolomic biomarkers affect meat quality.

The references are in the Aceduaco format.

Comments on the Quality of English Language

Minor editing of English language required

Author Response

Reviewer 1 Comments and Suggestions for Authors

The study is an interesting work on metabolomic biomarkers that influence the quality of various pig breeds. I believe that it has sufficient quality and scientific merit to be published in said journal, however, the authors must make some corrections in order to improve the manuscript.

Response:Thank you for reviewing our manuscript and valuable comments!

  1. Keywords must remove those that appear in the title such as: 'Shangai Local', 'Pig Breeds', LC-MS.

Response:Thank you for your comments. Based on your suggestion, we have removed the keywords 'Shangai Local', 'Pig Breeds', ‘LC-MS'. and replaced with “liquid chromatography-mass spectrometryï¼›longissimus dorsi muscleï¼›gluteus muscle” that I hope this is more appropriate.

  1. Introduction: in line 33, data must be entered worldwide and specifically in Shanghai on the production and consumption of this type of meat. In lines 42-43, the production levels or intervals of this type of compounds should be discussed in relation to the bromatological analysis: (ash, proteins, humidity, carbohydrates and fats). Some studies on the use of LC-MS for this type of matrices should be included in lines 46-47.

Response:Thank you for your comments. We reviewed the reference related to the pork consumption and added descriptions: “China is a major pork producer and consumer in the world, and it is reported that China's pork production has reached 52.96 million tons in 2020, accounting for more than 59.6% of meat production.”

  1. Materials and methods: Justify the experimental design used. Why are 96 samples used?

In the methodologies described for sections 2.3; 2.4 and 2.5 the authors must include the references of the methodologies they are using in case they are not theirs.

Response:Thank you for your comments. Generally, metabolomics analysis requires a minimum of 6 samples, we did 4 breeds, 2 tissues (L and T) for each breed, and 12 samples of each tissue, so the total was 4×2×12 = 96 samples. Based on your suggestions, we have added the references.

  1. Results and discussion: In Table 1, it would be interesting to perform a Pearson correlation between the variables. The figures have to be improved, since the resolution is not appropriate. In addition, the names of the metabolites must be standardized according to the IUPAC, in addition, some appear in capital letters and others in lower case.

Response:Thank you for your suggestion. We computed the Pearson correlation analysis on these meat traits (shown in the figure below). We added the corresponding result description “The Pearson correlation results on these meat traits demonstrated the protein content (%) showed a significant negative correlation with water content (%) (r = -0.51, P value = 1.59e-004. Meat lightness(L) trait significant positive correlation with the yellowness (b) trait (r= 0.62, P value = 1.785e-006), Protein content % (r=0.45, P value =0.001), and IMF% (r=0.44, P value =0.001).” If you feel the need to include the image in the results, we can place it in the article results or add it as a supplementary material.

        For the names of the metabolites according to the IUPAC issue. Metabolomics is a discipline that involves the quantitative analysis of all metabolites in living organisms, mostly small molecules with relative molecular mass of 1000 or less. These metabolites include various types of compounds, such as lipids, sugars, amino acids and various organic acids. Due to the variety and complex structure of metabolites, there is no uniform naming system that can cover all metabolites, and not every metabolite has an IUPAC standard name, so the metabolite names mentioned in this paper are derived from the English names included in the database. If wants to convert the names of the metabolites identified in this study to IUPAC names, need to check through the pubchem website: https://pubchem.ncbi.nlm.nih.gov/. In Supplementary Material S2, there is more detailed information on all differential metabolites and lipids, including chemical formular.

  1. Discussion: the authors should reinforce the discussion by justifying how metabolomic biomarkers affect meat quality.

Response:Thank you for your comments. We have added some reference reporting on its association with meat color traits such as: " The L-carnitine levels and its storage stabilities livestock products were reported have a high correlation (r=0.9764) with redness values of homogenized meat solution."  “Gao et al identified nine differential metabolites include L-carnitine were involved in antioxidant function, lipid metabolism, and cell signal transduction, which may decrease postmortem meat quality and play important roles in anti-heat stress.” Some of the other identified marker metabolites have indeed been less studied and the exact mechanism of action is still unclear, and future studies are needed.

  1. The references are in the Aceduaco format.

Response:Thank you for your comments. We have reorganized the reference and format based on the revised paper.

  1. Minor editing of English language required

Response:Thank you for your comments. We have made English language editing throughout the text in the hope that it is expressed more accurately. If language improvements are still needed, we are willing to accept the language change service suggested by the journal.

Reviewer 2 Report

Comments and Suggestions for Authors

This article provides useful information on the characterization of meat metabolites and lipids in local pig breeds in Shanghai, as determined by an LC-MS-based method. It is generally well designed and applied, but there are important points that need significant revision, with emphasis on reviewing the experimental design and presentation of results..

  1- In the abstract: Mention in passing how these specific metabolites and lipids might influence customer preference (taste, texture, etc.). 2- Line 17: "Better meat quality traits", Like what? 3- Please add the P. value to the results in the abstract. 4- Does the age of each breed differ? 5- How many samples are taken for analysis? 6- Please check that all abbreviations were provided in full names in their first mention.‎ 7- Line 38: "China being a center for domestication" is removed as it does not directly relate to the current study. 8- In the introduction: consider refining some of the cited references for better clarity and focus. 9- Lines 81-82: Authors should make it easier for the reader by adding the procedure or calculation method and the reference that was previously used or if it was modified. 10- Was blinding performed during sample collection or analysis to minimize bias? 11- Lines 72-73: "All slaughtering was done by licensed commercial slaughtering companies such as Wufeng Shangshi Food Co.,......" How? Is the slaughter process standardized for 4 breeds across all companies? 12- Line 84: (NY/T 821-84 2019) ? 13- If metabolites and lipids are analyzed. Why is this not expressed as a percentage of extract? 14- In Table 1: Unit "Shear force"? 15- Lines 117-119: Removal of features detected in less than 80% of samples. How? and explain this sentence. 16- Lines 121-122: Could be rephrased: "Differences between breed groups and muscle parts demonstrated by PLS-DA [17] and the permutation test was set to 200 times."  17- Briefly explain what OPLS-DA stands for and why it is used to assess model stability. 18- Briefly explain the filtering criteria and statistical tests used in the data analysis. 19- Add superscripts (e.g. a-c), the difference in means within rows for each breed compared to the commercial pig breed DLY is considered significant if p < 0.05 according to the "?" test. this test must determine the significant differences between breeds and thus on what basis the results were written. 20- However, for the purpose of this study, it was better to use the contrast or Dennett's tests instead of the " one-way ANOVA". 21- Rewrite and revise the results to make them clearer and check the statistical analysis as mentioned above. 22- P. Value in all manuscripts. 23- Reviewing the format of references and journal names, including abbreviations for each journal. 24- Lines 369-371 "Acknowledgments": it should be removed and replaced by a lines 374-375. 25- Please review the manuscript from the authors' point of view to simplify some complex sentences and to improve readability. 26- The conclusion could benefit from more detailed information on the observed effects.

Author Response

Reviewer 2 Comments and Suggestions for Authors

This article provides useful information on the characterization of meat metabolites and lipids in local pig breeds in Shanghai, as determined by an LC-MS-based method. It is generally well designed and applied, but there are important points that need significant revision, with emphasis on reviewing the experimental design and presentation of results.

Response:Thank you for reviewing our manuscript and for your valuable comments!

1- In the abstract: Mention in passing how these specific metabolites and lipids might influence customer preference (taste, texture, etc.).

Response:Thank you for your comments. We have added a sentence in the abstract “These specific metabolites and lipids might influence quality and taste properties, lead to customer preferences.”

2- Line 17: "Better meat quality traits", Like what?

Response:Thank you for your comments. Based on the results of our study (Table1), that Shanghai local pigs have higher IMF%, protein content %, and better meat color compared to commercial crossbred pig breed DLY, all of which indicate that the meat quality of Shanghai local pig breeds is better than that of DLY breed.

3- Please add the P. value to the results in the abstract.

Response: Thanks. We have added the P-value in the abstract.

4- Does the age of each breed differ?

Response: Yes. Since Shanghai local breeds grow slower than commercial crossbred pigs. This study was sampled on the basis of slaughter weight, the days of age for 100Kg weight of three Shanghai local pig breeds was at (290 ± 10 day) whereas the days of age for the same weight of pigs in DLY breed was at (170 ± 10 day).

5- How many samples are taken for analysis?

Response: We did 4 breeds, 2 tissues (L and T) for each breed, and 12 samples of each tissue, so the total was 4×2×12 = 96 samples for LC-MS analysis.

6- Please check that all abbreviations were provided in full names in their first mention.‎

Response:Thank you for your comments. We have addressed this issue.

7- Line 38: "China being a center for domestication" is removed as it does not directly relate to the current study.

Response:Thank you for your suggestion. We have removed this description and reference.

8- In the introduction: consider refining some of the cited references for better clarity and focus.

Response:Thank you for your comments. Following your suggestion, we have revised the Introduction to remove lines 54-58, which are not closely related to the focus of this paper.

9- Lines 81-82: Authors should make it easier for the reader by adding the procedure or calculation method and the reference that was previously used or if it was modified.

Response:Thank you for your comments. We added description for the method description and added reference: “Samples were taken from the thoracolumbar junction to the L muscle of the 3rd and 4th lumbar vertebrae for the measurement of meat quality traits.” Meat quality traits were determined with reference to the procedure of Agricultural industry standard of the People's Republic of China: technical code of practice for pork quality assessment (NY/T 821-84 2019) and reported methods.

10- Was blinding performed during sample collection or analysis to minimize bias?

Response:Thank you for your comments. We sampled randomly after slaughter based on individuals reaching the weight range and did not specifically try to screen the samples.

11- Lines 72-73: "All slaughtering was done by licensed commercial slaughtering companies such as Wufeng Shangshi Food Co.,......" How? Is the slaughter process standardized for 4 breeds across all companies?

Response:Thank you for your comments. Since slaughtering in China must be unified by licensed slaughtering company and private individuals are not allowed to engage in livestock slaughtering, the pigs in our experiment were slaughtered by slaughtering company, which were unified in accordance with industry standards. We sampled the specified parts meat for LC-MS experiment after their unified slaughter and division.

12- Line 84: (NY/T 821-84 2019) ?

Response:This is the document number of the Agricultural Industry Standard of the People's Republic of China, "Technical Practice for Pork Quality Assessment". Our meat quality traits in this study are measured by reference to this standardized method.

13- If metabolites and lipids are analyzed. Why is this not expressed as a percentage of extract?

Response:Thank you for your comments. LC-MS untargeted metabolomics is a relative quantitative method, the data provided is the peak area value surrounded by the chromatographic peaks and axes of metabolites, rather than the concentration value of metabolites. Different metabolites, due to the differences in their own physical and chemical properties, resulting in different instrumental response patterns, even if the same concentration of different metabolites, the signal intensity obtained from their detection will be different, or even a large difference, so it is not converted to a percentage. In addition, the untarget metabolome results focus on the relative abundance to compare the expression of the same metabolite in different samples or between different groups, and the conversion percentage is generally preferred to compare the expression of different metabolites in the same sample.

14- In Table 1: Unit "Shear force"?

Response:Thank you for pointing that out. The unit for Shear force is Kgf and we have modified it in the Table 1.

15- Lines 117-119: Removal of features detected in less than 80% of samples. How? and explain this sentence.

Response:Thank you for your comments. Raw data for untargeted metabolomics need to go through a series of pre-processing, mainly including missing value processing, normalization, QC quality control, data transformation, etc. In the description “Metabolic and lipidomic features detected at least 80 % in any set of samples were retained.”   In order to reduce the number of zeros present, Smilde et al. applied a procedure referred as the “80% rule”. In this work, the class information was utilized as supervisor, the ‘80% rule’ was modified to a ‘variable is kept if variable has a non-zero value for at least 80% in the samples of any one class’. Which called as modified 80% rule. We have added reference accordingly.

Reference:

Smilde, Age K., et al. "ANOVA-simultaneous component analysis (ASCA): a new tool for analyzing designed metabolomics data." Bioinformatics 21.13 (2005): 3043-3048.

Hrydziuszko O, Viant M R. Missing values in mass spectrometry based metabolomics: an undervalued step in the data processing pipeline[J]. Metabolomics, 2012, 8(s1):161-174.

Yang J, Zhao X, Lu X, et al. A data preprocessing strategy for metabolomics to reduce the mask effect in data analysis[J]. Frontiers in Molecular Biosciences, 2015: 4-4.

16- Lines 121-122: Could be rephrased: "Differences between breed groups and muscle parts demonstrated by PLS-DA [17] and the permutation test was set to 200 times." 

Response:Thank you for your comments. We modified it with “Perform variance analysis on the matrix file after data preprocessing. Overall differences among the groups were analyzed by PLS-DA (Partial Least Squares Discriminant Analysis). The number of times a randomized permutation test was set to 200.”

17- Briefly explain what OPLS-DA stands for and why it is used to assess model stability.

Response:Thank you for your comments. OPLS-DA (Orthogonal Partial Least Squares Discriminant Analysis) is a derivative algorithm of PLS-DA. Compared with PLS-DA, OPLS-DA is a combination of Orthogonal Signal Correction (OSC) and PLS-DA. In contrast to PLS, OPLS has the effect of filtering out/ignoring "noise" variables in the matrix of observed variables X that are not correlated with the matrix of predictive variables Y, i.e., it removes uncorrelated or orthogonal variation in the data variables of X that are not correlated with the variables of Y. OPLS-DA is able to better differentiate between subgroups and improve the model's validity and parsing ability.

18- Briefly explain the filtering criteria and statistical tests used in the data analysis.

Response:Thank you for your comments. The data processing process of this study was analyzed with reference to a standardized process (analyzed through the free online platform (Cloud.majorbio.com) (Ren, et al., 2022). Its entire process is shown in the figure below and referenced to the following references. We have revised the section on data processing and added the corresponding references.

Reference:

Karaman I. Preprocessing and Pretreatment of Metabolomics Data for Statistical Analysis[M]// Metabolomics: From Fundamentals to Clinical Applications. Springer International Publishing, 2017.

Hrydziuszko O, Viant M R. Missing values in mass spectrometry based metabolomics: an undervalued step in the data processing pipeline[J]. Metabolomics, 2012, 8(s1):161-174.

Yang J, Zhao X, Lu X, et al. A data preprocessing strategy for metabolomics to reduce the mask effect in data analysis[J]. Frontiers in Molecular Biosciences, 2015: 4-4.

19- Add superscripts (e.g. a-c), the difference in means within rows for each breed compared to the commercial pig breed DLY is considered significant if p < 0.05 according to the "?" test. this test must determine the significant differences between breeds and thus on what basis the results were written.

Response:Thank you for your comments. We applied one-way ANOVA (between multiple groups) and then post-hoc tests compared two-by-two to detect sample groups that differed in multiple groups. Student’s t-test (unpaired) was applied to identified DEMs between two groups.

20- However, for the purpose of this study, it was better to use the contrast or Dennett's tests instead of the "one-way ANOVA".

Response:Thank you for your comments. We may have been less precise in our description in the abstract and results, so we rechecked the results for relevant significance, such as the 2 key metabolites mentioned in the abstract, and their significance results in these three local pig breeds compared to the DLY pig breeds are as follows. We have revised the description of the relevant results accordingly.

L-Carnitine

DLY_L_vs_MSZ_L

DLY_L_vs_SWT_L

DLY_L_vs_SHB_L

P-Value 

0.0006355

2.78E-05

0.8603

DLY_T_vs_MSZ_T

DLY_T_vs_SWT_T

DLY_T_vs_SHB_T

P-Value 

8.30E-07

5.75E-08

0.04627

PA(18:0/18:2)

DLY_L_vs_MSZ_L

DLY_L_vs_SWT_L

DLY_L_vs_SHB_L

 P-Value

0.0004586

5.04E-06

0.1446

DLY_T_vs_MSZ_T

DLY_T_vs_SWT_T

DLY_T_vs_SHB_T

P-Value  

0.000103

2.25E-08

3.07E-07

21- Rewrite and revise the results to make them clearer and check the statistical analysis as mentioned above.

Response: Thank you for your comments.  We have checked results and modified description accordingly.

22- P. Value in all manuscripts.

Response:Thanks. We modified and uniformly write P-values in the manuscripts.

23- Reviewing the format of references and journal names, including abbreviations for each journal.

Response:Thanks. We have checked and modified in all manuscripts.

24- Lines 369-371 "Acknowledgments": it should be removed and replaced by a lines 374-375.

Response:Thank you for pointing out the error, we have corrected it accordingly.

25- Please review the manuscript from the authors' point of view to simplify some complex sentences and to improve readability.

Response:Thanks. We have revised the entire text in the light of the problems you have pointed out, and we hope to make it clearer.

26- The conclusion could benefit from more detailed information on the observed effects.

Response:Thank you for your suggestion. We have modified conclusion section and added more detailed information on the observed effects.
